# Synthesis and Structural Characterization of a Silver(I) Pyrazolato Coordination Polymer [note 1]

**DOI:** 10.3390/molecules26041015

**Published:** 2021-02-15

**Authors:** Kiyoshi Fujisawa, Takuya Nemoto, Yui Morishima, Daniel B. Leznoff

**Affiliations:** 1Department of Chemistry, Ibaraki University, Mito, Ibaraki 310-8512, Japan; cu_peroxo@yahoo.co.jp (T.N.); fujisawa0608@gmail.com (Y.M.); 2Department of Chemistry, Simon Fraser University, 8888 University Drive, Burnaby, BC V5A1S6, Canada

**Keywords:** polynuclear, silver, crystal structure, pyrazolate ligand, coordination polymer

## Abstract

Coinage metal(I)···metal(I) interactions are widely of interest in fields such as supramolecular assembly and unique luminescent properties, etc. Only two types of polynuclear silver(I) pyrazolato complexes have been reported, however, and no detailed spectroscopic characterizations have been reported. An unexpected synthetic method yielded a polynuclear silver(I) complex **[Ag(μ-L1Clpz)]*_n_*** (L1Clpz^−^ = 4-chloride-3,5-diisopropyl-1-pyrazolate anion) by the reaction of {[Ag(μ-L1Clpz)]_3_}_2_ with (^n^Bu_4_N)[Ag(CN)_2_]. The obtained structure was compared with the known hexanuclear silver(I) complex {[Ag(μ-L1Clpz)]_3_}_2_. The Ag···Ag distances in **[Ag(μ-L1Clpz)]*_n_*** are slightly shorter than twice Bondi’s van der Waals radius, indicating some Ag···Ag argentophilic interactions. Two Ag–N distances in **[Ag(μ-L1Clpz)]*_n_*** were found: 2.0760(13) and 2.0716(13) Å, and their N–Ag–N bond angles of 180.00(7)° and 179.83(5)° indicate that each silver(I) ion is coordinated by two pyrazolyl nitrogen atoms with an almost linear coordination. Every five pyrazoles point in the same direction to form a 1-D zig-zag structure. Some spectroscopic properties of **[Ag(μ-L1Clpz)]*_n_*** in the solid-state are different from those of {[Ag(μ-L1Clpz)]_3_}_2_ (especially in the absorption and emission spectra), presumably attributable to this zig-zag structure having longer but differently arranged intramolecular Ag···Ag interactions of 3.39171(17) Å. This result clearly demonstrates the different physicochemical properties in the solid-state between 1-D coordination polymer and metalacyclic trinuclear (hexanuclear) or tetranuclear silver(I) pyrazolate complexes.

## 1. Introduction

Cyclic trinuclear complexes with monovalent coinage metal ions have been of interest to coordination chemists for three decades [1,2,3]. One of the ligands known to form these cyclic trinuclear complexes is pyrazolate [4,5,6], which is known to act as an A-frame-like bridging ligand with some metal ions, with an Npz–M–Npz linear coordination mode (pz = pyrazolate anion, C_3_H_3_N_2_^−^) [1,2,3,4,6,7,8,9]. Early studies in 1970 suggested that silver(I) pyrazolato complexes existed as a polymeric 1-D chain [Ag(pz)]*_n_* (Figure 1, left) [10]. The structure of [Ag(pz)]*_n_* was discussed in that the deprotonated pyrazolato complexes are “at least trimeric but polymeric forms cannot be excluded”, based on far-IR spectroscopy data [11]. The first structural characterization of pyrazolato complexes coordinated by coinage metal(I) ions was determined by Fackler and co-workers in 1988: their reported silver(I) pyrazolato complex had three intramolecular Ag···Ag distances of 3.306(2), 3.362(2), and 3.496(2) Å, forming a trinuclear complex [Ag(3,5-Ph_2_pz)_3_] (3,5-Ph_2_pz = 3,5-diphenyl-1-pyrazolate anion) [12]. After this report, the hexanuclear complex {[Ag(3,5-Ph_2_pz)_3_]}_2_ was also reported, having one intermolecular Ag···Ag interaction of 2.9712(14) Å without any crystalline solvents [13]. This dimerization to build the hexanuclear structure can be caused by an additional stabilization of the silver(I) ions provided by argentophilic interactions (Ag···Ag interactions) [14]. Subsequent ab initio powder X-ray diffraction (XRD) evidence also indicated two possibilities: silver(I) pyrazolato complexes could exist as either a coordination polymer [Ag(pz)]*_n_* with an intramolecular Ag···Ag distance of 3.40 Å or as a dimeric trinuclear {[Ag(pz)]_3_}_2_ structure with an intermolecular Ag···Ag distance of 3.431(2) Å, depending on the method of synthesis [15]. After that, the single-crystal characterization of the product made by the same synthetic method with aqueous NH_3_ was reported as [Ag(pz)]*_n_* with an intramolecular Ag···Ag (argentophilic) interaction of 3.3718(7) Å, an intermolecular Ag···Ag interaction of 3.2547(6) Å, and an N–Ag–N angle of 169.98° (14) (Appendix A) [16]. However, its physicochemical properties such as solid-state photoluminescence have not been reported [15,16].

We have previously reported silver(I) pyrazolato complexes synthesized by alkyl- and aryl-substituted pyrazoles to form trinuclear and tetranuclear structures, depending on the method of synthesis and the nature of the substituents on the pyrazolate ring (Figure 1, center and right) [17,18,19,20,21,22]. Some Au···Ag (metallophilic) interactions in cyclic trinuclear coinage metal(I) complexes were found [1,2]. Au-Im complexes can be easily interacted with silver(I) ions to form Au···Ag interactions, since the order of π-acidity is Au < Cu < Ag for a given ligand, and Im^−^ (imidazolate) < Pyridine < Cb^−^ (carbeniate) < Pz^−^ for a given metal [1,2,3]. As an example, the mixed metal(I) complex [(Ag{([Au(*C*^2^,*N*^3^-bzim)]_3_)_2_}](BF_4_) with Au···Ag interactions was reported (bzim^−^ = 1-benzylimidazole anion) [23].

With this electronic preference in mind, to construct Ag···Ag interactions by silver(I) pyrazolato complexes, it should be valuable to explore the different strategy of using electro-withdrawing substituents on the pyrazole ligand and [Ag(CN)_2_]^−^ as a silver(I) ion source. The dicyanoargentate(I) anion is very useful to build coordination polymers [24,25,26,27,28] and performs well as a building block in general [29]. In this article, we report the detailed structure and characterizations as well as the unexpected synthetic procedure that leads to the new polynuclear silver(I) pyrazolato complex **[Ag(μ-L1Clpz)]*_n_*** (L1Clpz^−^ = 4-chloride-3,5-diisopropyl-1-pyrazolate anion) (Figure 2).

## 2. Results and Discussion

### 2.1. Synthesis

The reaction of {[Ag(μ-L1Clpz)]_3_}_2_ with 1.5 equivalents of (^n^Bu_4_N)[Ag(CN)_2_] in ethyl acetate was carried out at room temperature as shown in Scheme 1. After one week of reaction time, a white powder was gradually generated. Its IR spectroscopic measurement revealed that the ν(C≡N) stretching peak of the starting material (^n^Bu_4_N)[Ag(CN)_2_] at 2141 cm^−1^ disappeared and a new complex that did not contain any cyanide was generated. From a slow liquid-liquid diffusion method, we obtained a few single-crystals to reveal the structure (see Section 2.2). The above white powders prepared in the bulk synthetic method were the same as that obtained in single-crystal form and structurally characterized (below), as confirmed by the comparison of the powder X-ray diffractogram of the powder and the powder pattern generated from the single-crystal structure, as shown in Appendix A. As mentioned in the Introduction, although a [Ag(μ-L1Clpz)]_3_/[Ag(CN)_2_]^−^ based polynuclear structure with intermolecular Ag···Ag interactions was initially targeted, only the polynuclear silver(I) complex **[Ag(μ-L1Clpz)]_n_**, identified as a coordination polymer was obtained (Scheme 1); the detailed mechanism leading to this product was not explored. This coordination polymer assembly was only successfully performed by {[Ag(μ-L1Clpz)]_3_}_2_ but was not observed either by the non-halogenated derivative {[Ag(μ-L1pz)]_3_}_2_ or the other halogenated derivatives {[Ag(μ-L1Brpz)]_3_}_2_ and {[Ag(μ-L1Ipz)]_3_}_2_ (Figure 2). The reactions of these other hexanuclear complexes with (^n^Bu_4_N)[Ag(CN)_2_] did not produce any powder precipitate. Indeed, after all of the solvent had slowly evaporated from the reaction mixture, powder XRD measurements were carried out on the resulting residues, which indicated that no new complexes were generated: only a physical mixture of the initial starting materials, i.e., the hexanuclear complexes {[Ag(L1Xpz)]_3_}_2_ (X = H, Br, and I) and (^n^Bu_4_N)[Ag(CN)_2_] were observed. We also tried to make polynuclear [Ag(μ-L1Xpz)]_*n*_ (X = H, Cl, Br, and I) by the literature method by using aqueous NH_3_ [15,16], but only the same hexanuclear complexes {[Ag(L1Xpz)]_3_}_2_ (X = H, Cl, Br, and I) were obtained.

### 2.2. Structure

Single-crystal X-ray structural analysis was performed on the polynuclear silver(I) material **[Ag(μ-L1Clpz)]*_n_***, the perspective drawing of which is shown in Figure 3. The relevant bond lengths (Å) and angles (°) are noted in the caption. The 1-D coordination polymer **[Ag(μ-L1Clpz)]*_n_*** and packing diagram are drawn in Figure 4 and Appendix A, respectively.

**[Ag(μ-L1Clpz)]*_n_*** exists as a coordination polymer with an intramolecular argentophilic distance of Ag1···Ag2, 3.39171(17) Å, which is slightly shorter than twice Bondi’s van der Waals radius (3.44 Å = 1.72 Å × 2) [30], indicating the presence of Ag···Ag argentophilic interactions [14]. Two Ag–N distances were found: Ag1–N1, 2.0760(13) Å and Ag2–N2, 2.0716(13) Å. The N–Ag–N bond angles of 180.00(7)° (N1–Ag1–N1′) and 179.83(5)° (N2′–Ag2′–N2”) indicate that each silver(I) ion has two pyrazolyl nitrogen atoms coordinated in an almost linear fashion. Two of the pyrazolate aromatic rings are completely co-planar with a dihedral angle of 0° (between pyrazole ring 1 and 2), and another pair is twisted relative to each other, with a dihedral angle of 79.24(7)° (between pyrazole ring 2 and 3) as shown in Figure 3. Every five pyrazole units point in the same direction, thereby forming a 1-D zig-zag structure. Each 1-D zig-zag structure is isolated and there are no significant interactions with the neighboring zig-zag chain. Within each 1-D zig-zag structure, two additional Ag···Ag distances are noteworthy: Ag1···Ag1#, 4.73335(10) Å and Ag2···Ag2′, 6.7834(3) Å. Moreover, this zig-zag structure has two Ag-based different angles with Ag2···Ag1···Ag2′, 180.000(4)° and Ag1···Ag2′···Ag1#, 88.499(6)° (Figure 3).

This **[Ag(μ-L1Clpz)]*_n_*** structure is completely different from the reported one for [Ag(pz)]*_n_* which exhibits an intramolecular Ag···Ag distance of 3.3718(7) Å. The zig-zag chain in [Ag(pz)]*_n_* also interacts with neighboring zig-zag chains via inter-chain Ag···Ag distances of 3.2547(6) Å (Appendix A). The zig-zag structure in **[Ag(μ-L1Clpz)]*_n_*** is also different from the original hexanuclear silver(I) complex {[Ag(μ-L1Clpz)]_3_}_2_, which features a dimeric trinuclear structure with three intermolecular argentophilic interactions of 3.1003(17), 3.1298(15), and 3.1051 Å to form an overall hexanuclear structure (Appendix A) [20]. These distances are significantly shorter than twice Bondi’s van der Waals radius (3.44 Å) [30], indicating a strong Ag···Ag argentophilic interaction [14] (Appendix A). The layered structure of {[Ag(μ-L1Clpz)]_3_}_2_ did not interact with the neighboring layered one (Appendix A). The layered structure was formed with longer Ag···Ag distances between two {[Ag(μ-L1Clpz)]_3_}_2_ complexes of 7.3531(18), 7.3881(17), and 7.5258(18) Å.

### 2.3. Solution-State Properties

The ^1^H-NMR spectrum of the obtained white powder **[Ag(μ-L1Clpz)]*_n_*** in CDCl_3_ was measured (Appendix A) and the observed chemical shifts are identical to those of {[Ag(μ-L1Clpz)]_3_}_2_ (Appendix A) [20], indicating that the supramolecular solid-state 1-D zig-zag structure of **[Ag(μ-L1Clpz)]*_n_*** is not stable upon dissolution, converting to form the known hexanuclear silver(I) complex {[Ag(μ-L1Clpz)]_3_}_2_. This observation is also supported by its solution-state UV-Vis spectrum in cyclohexane (Appendix A) and photoluminescence spectrum in cyclohexane (Appendix A). The maximum peak position in UV-Vis spectrum is 226 nm which is the same position as that of {[Ag(μ-L1Clpz)]_3_}_2_ (226 nm) [20]. Moreover, the emissive maximum is 307 nm (280 nm excitation), which is also the same position as that of {[Ag(μ-L1Clpz)]_3_}_2_. Therefore, detailed comparisons between **[Ag(μ-L1Clpz)]*_n_*** and {[Ag(μ-L1Clpz)]_3_}_2_ were carried out by solid-state spectroscopies.

### 2.4. Solid-State Properties

IR and Raman spectra of the **[Ag(μ-L1Clpz)]*_n_*** and {[Ag(μ-L1Clpz)]_3_}_2_ complexes are reproduced in Appendix A, respectively. The C=N stretching vibrations of 1506 cm^−1^ (IR) and of 1507 and 1495 cm^−1^ (Raman) in **[Ag(μ-L1Clpz)]*_n_*** are almost the same as the C=N stretching vibrations of 1505 cm^−1^ (IR) and 1495 cm^−1^ (Raman) in {[Ag(μ-L1Clpz)]_3_}_2_. The C–Cl stretching vibrations could be observed in the far-IR region of 587 cm^−1^ (IR) and 575 cm^−1^ (Raman) in **[Ag(μ-L1Clpz)]*_n_***, which are slightly shifted from 579 cm^−1^ (IR) and 573 cm^−1^ (Raman) in {[Ag(μ-L1Clpz)]_3_}_2_ [20], respectively. The Ag–N stretching vibration should be observed at around 510 cm^−1^ [20,21,31] and was observed at 511 cm^−1^ (Raman) in **[Ag(μ-L1Clpz)]*_n_***, which was the same energy at 511 cm^−1^ (Raman) in {[Ag(μ-L1Clpz)]_3_}_2_ [20]. These vibrational spectroscopy comparisons indicate that each stretching vibration energy is almost the same as the other, although there is a clearly measurable difference in the C-Cl stretches, consistent with the presence of intercluster Cl···Cl interactions in the hexanuclear system (3.852 Å) that are absent in the 1-D zig-zag chain structure.

The solid-state UV-Vis absorption spectrum of **[Ag(μ-L1Clpz)]*_n_*** acquired as a Nujol suspension is shown in Figure 5, along with that of {[Ag(μ-L1Clpz)]_3_}_2_ [20] for comparison. The characteristic absorption band at 225 nm in {[Ag(μ-L1Clpz)]_3_}_2_ [20] was obviously shifted to lower energy at 248 nm with a shoulder peak around 280 nm for the 1-D zig-zag structure. This is clearly different from the behavior of solution-state UV-Vis absorption spectra as shown in Appendix A. This shift in **[Ag(μ-L1Clpz)]*_n_*** may be caused by polynuclear formation with intramolecular argentophilic interactions of Ag1···Ag2, 3.39171(17) Å as shown in Figure 3. For this detailed assignment, density functional theory calculations are required but are beyond the scope of this article. Nevertheless, the absorption band can be assigned to a silver(I) to pyrazolate change transfer (MLCT) based on the other reported hexanuclear coinage metal(I) complexes [18,19,20,32].

The emission spectrum of **[Ag(μ-L1Clpz)]*_n_*** in the solid-state was also somewhat different from that of the hexanuclear silver(I) analogue {[Ag(μ-L1Clpz)]_3_}_2_ [20] as shown in Figure 6 at room temperature, recorded using a 280 nm excitation wavelength. In {[Ag(μ-L1Clpz)]_3_}_2_, the main emissive band at 374 nm and an additional small one at 312 nm were observed. However, the emission of the coordination polymer **[Ag(μ-L1Clpz)]*_n_*** was clearly shifted to higher energy around 314 nm, and a new broad peak around 490 nm was observed. 

The temperature-dependent photoluminescence spectra in **[Ag(μ-L1Clpz)]*_n_*** were recorded (Figure 7). The corresponding variable temperature emission spectra for {[Ag(μ-L1Clpz)]_3_}_2_ are also shown in Appendix A [20]. The more intense 473 nm emission band of **[Ag(μ-L1Clpz)]*_n_*** at 83 K exhibited an additional vibrational fine structure around 346 nm, which was observed only at 83 K. From this vibrational behavior, this higher energy emission may be from ligand-based phosphorescence. On the other hand, the lower energy emission band was attributed to metal-based phosphorescence arising from closed shell d^10^–d^10^ intramolecular Ag···Ag interactions [18,19,20,21,32,33,34]. For the original hexanuclear complex {[Ag(μ-L1Clpz)]_3_}_2_, the intensity of both bands at 374 and 312 nm gradually increased as the measurement temperature decreased. No bands around 470 nm were observed, even at low temperature. From these observations, the lower energy strong emission band at 473 nm in **[Ag(μ-L1Clpz)]*_n_*** is very unique due to its polynuclear supramolecular structure. We are now in the process of probing the origin of this stimulating behavior by theoretical and more detailed physicochemical researches.

## 3. Materials and Methods

### 3.1. Material and General Techniques

The preparation and handling of all complexes was performed under an argon atmosphere using standard Schlenk tube techniques. Ultra-dry ethyl acetate was purchased from Wako Pure Chemical Ind. Ltd. and deoxygenated by purging with argon gas. Deuteriochloroform was obtained from Cambridge Isotope Laboratories, Inc. Other reagents were commercially available and were used without further purification. The 3,5-diisopropyl-1-pyrazole (L1pz-H) [35] and its halogenated pyrazoles (L1Clpz-H, L1Brpz-H, and L1Ipz-H) [20] were prepared by published methods. (^n^Bu_4_N)[Ag(CN)_2_] was obtained by the reaction of KAg(CN)_2_ (1.016 g, 5.10 mmol) and ^n^Bu_4_NBr (1.7029 g, 5.28 mmol) in 20 mL of H_2_O at room temperature for 2 hours to form a white powder of (^n^Bu_4_N)[Ag(CN)_2_] (1.801 g, 4.48 mmol, 88% yield). Hexanuclear silver(I) complexes ({[Ag(μ-L1Clpz)]_3_}_2_, {[Ag(μ-L1Brpz)]_3_}_2_, and {[Ag(μ-L1Ipz)]_3_}_2_) were obtained by the published methods [20].

### 3.2. Instrumentation

IR spectra (4000–400 cm^−1^) and far-IR spectra (680–150 cm^−1^) were recorded as KBr pellets using a JASCO FT/IR-6300 spectrophotometer and as CsI pellets using a JASCO FT/IR 6700 spectrophotometer (JASCO, Tokyo, Japan), respectively. Raman spectra (4000–200 cm^−1^) were measured as powders on a JASCO RFT600 spectrophotometer with a YAG laser 600 mW (JASCO, Tokyo, Japan). Abbreviations used in the description of vibration data are as follows: s, strong; m, medium; w, weak. ^1^H-NMR (500 MHz) spectra were obtained on a Bruker AVANCE III-500 NMR spectrometer at room temperature (298 K) in CDCl_3_ (Bruker, Yokohama, Japan). ^1^H chemical shifts were reported as *δ* values relative to residual solvent peaks. UV-Vis spectra (solution and solid, 800–200 nm) were recorded on a JASCO V-570 spectrophotometer (JASCO, Tokyo, Japan). The values of *ε* were calculated per metal(I) ion. Solid samples (mulls) for spectroscopy were prepared by finely grinding microcrystalline material into powders with a mortar and pestle and then adding mulling agents (Nujol, poly(dimethylsiloxane), viscosity 10,000 (Aldrich)) before uniformly spreading it between quartz plates. Powder X-ray diffraction (XRD) measurements were conducted on a Rigaku SmartLab-SP/IUA X-ray diffractometer (Rigaku, Tokyo, Japan) with a Cu Kα radiation (λ = 1.54 Å) source (40 kV, 30 mA) and a high-speed one-dimensional detector D/teX Ultra 250. The 2*θ* was measured in the range of 5–90° with a scan step of 0.02° and scan speed of 10° min^−1^. Solid samples for XRD were prepared by finely grinding microcrystalline materials into powders with a mortar and pestle and then placing them on an aluminum dish (0.2 mm thickness). Luminescence spectra were recorded on a JASCO FP-6500 (solid, 700–300 nm) spectrofluorometer (JASCO, Tokyo, Japan). Low-temperature electronic absorption and luminescence spectra were recorded using solid samples cooled with a liquid nitrogen cryostat (CoolSpeK USP-203) from Unisoku Scientific Instruments (Osaka, Japan). The elemental analyses (C, H, and N) were performed by the Chemical Analysis Center of Ibaraki University. 

### 3.3. Preparation of Complexes

#### **[Ag(μ-L1Clpz)]_n_** 

{[Ag(μ-L1Clpz)]_3_}_2_ (54.5 mg, 0.0309 mmol) and (^n^Bu_4_N)[Ag(CN)_2_] (19.1 mg, 0.0475 mmol) were dissolved in ethyl acetate (6 cm^3^) and the solution was stirred at room temperature for one week. A white solid was precipitated during this time. After this, the white solid was filtered and washed with a small amount of ethyl acetate. The obtained white powder as **[Ag(μ-L1Clpz)]*_n_*** was dried by vacuum pump.

Yield: 22.7 mg, 0.077 mmol, 42%.

Calcd for C_9_H_14_AgClN_2_·1/4(H_2_O): C, 36.27; H, 4.90; N, 9.40. Found: C, 36.21; H, 4.57; N, 9.20. IR (KBr, cm^−1^): 2971s(C-H), 2925s(C-H), 2862s(C-H), 1506m(C≡N), 1461m, 1400m, 1367m, 1361m, 1283m, 1145s, 1118s, 1092s, 1051s, 1035s, 804w. Far-IR(CsI, cm^−1^): 685w, 587m(C-Cl), 561w, 540w, 501,476w, 437w, 411w, 389w, 371w, 349w, 276w, 237br. Raman (solid, cm^−1^): 2974s(C-H), 2913s(C-H), 2865s(C-H), 1507m(C≡N), 1495m(C≡N), 1461m, 1448m, 1430m, 1382m, 1369s, 1300m, 1283m, 1176w, 1145w, 1107m, 957w, 880m, 706w, 657m, 575m(C-Cl), 511m(Ag-N), 438w, 385w, 341w, 268m. ^1^H-NMR (CDCl_3_, 500 MHz): δ 1.41 (d, *J* = 7 Hz, 36H, CH*Me*_2_), δ 3.13 (sept, *J* = 7 Hz, 6H, C*H*Me_2_). UV-Vis (solution, cyclohexane, λ_max_/nm(ε/cm^−1^ mol^−1^ dm^3^)) 226 (39200). UV-Vis (solid, Nujol, nm): 248, 280 (sh). Emission at 280 nm excitation wavelength (solution, cyclohexane, λ_max_/nm): 307. Emission at 280 nm excitation wavelength (solid, λ_max_/nm): 299 K, 315, 490; 173 K, 314, 474, 614sh; 83 K, 317, 331, 346, 362, 379, 456sh, 473.

Liquid-liquid diffusion was applied to obtain crystals at room temperature. The {[Ag(μ-L1Clpz)]_3_}_2_ (0.05 mmol) in 10mL of ethyl acetate was transferred to a 30 mL Erlenmeyer flask. On this solution, the solution containing (^n^Bu_4_N)[Ag(CN)_2_] (0.1 mmol) dissolved in 10 mL of ethyl acetate was carefully layered and then the top was covered by parafilm. After a few weeks, some crystals were formed, which were suitable for single-crystal X-ray analysis.

### 3.4. X-ray Crystal Structure Determination

Crystal data and refinement parameters for the silver(I) pyrazolato coordination polymer **[Ag(μ-L1Clpz)]*_n_*** are given in Table 1. All crystallographic data have been deposited at the Cambridge crystallographic data center (CCDC), 12 Union Road, Cambridge CB2 1EZ, UK and copies can be obtained on request, free of charge, by quoting the publication citation and the deposition number. CCDC number: 2053891.

The diffraction data were measured on a Rigaku/MSC Mercury CCD system (Rigaku, Tokyo, Japan) with graphite monochromated Mo Kα (λ = 0.71070 Å) radiation at −80 °C. The unit cell parameters of each crystal were determined using CrystalClear [36] from 6 images. The crystal to detector distance was 44.74 mm. Data were collected using 0.5° intervals in φ and ω to a maximum 2*θ* value of 55.0°. A total of 744 oscillation images were collected. The highly redundant data sets were reduced using CrysAlisPro [37]. An empirical absorption correction was applied for each complex. Structures were solved by direct methods (SIR2008) [38]. The position of the silver ions and their first coordination sphere were located from a direct method *E*-map; other non-hydrogen atoms were found in alternating difference Fourier syntheses and least squares refinement cycles. During the final refinement cycles the temperature factors were refined anisotropically. Refinement was carried out by a full matrix least-squares method on *F*^2^. All calculations were performed with the CrystalStructure [39] crystallographic software package except for refinement, which was performed using SHELXL 2013 [40]. Hydrogen atoms were placed in calculated positions. Crystallographic data and structure refinement parameters including the final discrepancies (*R* and *R*w) are listed in Table 1.

## 4. Conclusions

By using an unexpected synthetic method, we obtained a silver(I) pyrazolato complex **[Ag(μ-L1Clpz)]*_n_*** as a coordination polymer by the reaction of {[Ag(μ-L1Clpz)]_3_}_2_ with (^n^Bu_4_N)[Ag(CN)_2_]. This polynuclear silver(I) structure was compared with the known hexanuclear silver(I) structure {[Ag(μ-L1Clpz)]_3_}_2_. Two N–Ag–N bond angles of 180.00(7)° and 179.83(5)° in the silver(I) coordination polymer **[Ag(μ-L1Clpz)]*_n_*** indicate that each silver(I) ion is coordinated by two pyrazolyl nitrogen atoms with an almost linear coordination. Every five pyrazoles point in the same direction to form a 1-D zig-zag structure. This zig-zag structure is not stable in solution, but it converts to the original hexanuclear silver(I) complex {[Ag(μ-L1Clpz)]_3_}_2_. In the solid-state photoluminescence spectrum, a lower energy strong emission band at 473 nm at lower temperature is very unique and attributable to the differences in polynuclear structure between the two systems. Silver(I) complexes are not generally so emissive, even at lower temperature and thus this 1-D zig-zag polynuclear structure is particularly noteworthy in coinage metal(I) pyrazolate research. This **[Ag(μ-L1Clpz)]*_n_*** broadens a family of silver(I) coordination polymers [24,25,41,42,43]. Moreover, some high antibacterial activity research using silver(I) coordination polymers are also reported [41,42,43,44]. Further efforts to probe how the structure of coinage metal(I) pyrazolates is affected by ligand and coordination environment are in progress. 

## Data Availability

Data is contained within the article and supplementary material.

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
