# Peer review of "Synthesis and Structural Characterization of a Silver(I) Pyrazolato Coordination Polymer†"

_molecules, 2021, doi:10.3390/molecules26041015_

Round 1
Reviewer 1 Report
The manuscript by K. Fujisawa et al. reports the synthesis and characterization of a 1D polymeric Ag(I) pyrazolate. The new material was characterized by an elemental analysis, IR, Raman, UV-vis and 1H-NMR spectroscopies and its crystal structure was determined by single crystal X-ray diffraction methods. The solid state emission spectra are also reported, highlighting the difference from the emission spectra of dimer-of-Ag3 trimer assemblies. The new complex is well characterized and adequately described. This manuscript follows up on an earlier publication by the same group (ref 20). It can be accepted for publication with the following minor comments:
Abstract, 2nd line: The authors probably mean to say “Only two types of polynuclear silver(I)…”, as several structures of silver(I) pyrazolates have been published.
Introduction, lines 40-42: The Ag…Ag distances mentioned (from ref-13) are intramolecular (not intermolecular) and the Ag3 complex reported in ref-13 is a distinct molecule with no short intermolecular contacts. The authors have possibly in mind the (Ag3)2 structure reported also by Fackler et al. in Inorg. Chim. Acta 2005, 358, 1657, or the ones in Cryst. Growth Des. 2013, 13, 264. In the same sentence, ref-12 reported only a Cu3 species, not relevant to the Ag3 complexes.
Page 2, line 47 and page 3, line 117: intramolecular, not intermolecular.
Page 2, line 47: Besides ref-21, tetranuclear structures have also been reported in Inorg. Chim. Acta 2007, 360, 2503.
Page 4, lines 127, 128, 130 and 135: It is not correct to describe this structure as spiral. A spiral structure should be chiral, which this 1-D structure is not, because it changes the direction of its rotation as it propagates. It is better described as a zig-zag structure, but with different dihedral and torsion angles than the one reported in ref-15.
Reviewer 2 Report
There is no flaw in crystal structure and the article is presented well. But Molecule's impact factor is about 3.27 and work in the paper is not in that level, I guess. Pyrazolato ligands are known already and they are NOT new. Complexation process is also simple. Moreover, there is not much application-oriented studies on the silver complex except photo-physical properties at lower temperature. To me this paper should be published in the journals such as Polyhedron or Inorganica chimica Acta. My recommendation is declined for publication.
Author Response
We are really very disappointed with this comment and obviously do not agree with this assessment. This chemistry is very important, with a wide diversity of structures with pyrazolates reported and widely quoted in the literature, as shown in Figure 1. However, despite this, polynuclear structures are very limited: only two structures have been reported and even in those rare cases their chemical and physical properties have not been reported. In this manuscript, for the first time, the detailed properties in solid-state are now reported with respect to a new 1-D pyrazolate-based structure, which we are confident will be noticed by the inorganic and coordination chemistry community in particular. Thus, we believe that this manuscript has the merits to be published in Molecules, as is clearly supported by the other two referees.
Reviewer 3 Report
This interesting work reports on the synthesis and full characterization of a new silver pyrazolato 1D coordination polymer obtained via an unexpected synthetic procedure. The crystal structure of the obtained compound was discussed in detail with a focus on argentofilic interactions. Spectroscopic features and luminescent behavior of the obtained coordination polymer were also explored. The results are very interesting and widen the family of silver coordination polymers. The study is novel, technically correct, well supported by supplementary materials, and fits the scope of the journal. Its publication is thus recommended.
The following minor points should be addressed in a revised version.
1. The compound is in fact a coordination polymer. I therefore suggest to use this name "coordination polymer" instead of "complex" in the title and text.
2. If possible, I suggest to add structural formula of the obtained compound in Scheme 1. A style of Cambridge Structural Database can be used.
3. In section 3.3, please briefly comment on a solubility of the obtained silver compound (this might be important for exploring its bioactivity in future studies).
4. Conclusions. It would be good to add a comment that "the obtained compound broadens a family of Ag(I) coordination polymers" with interesting functional properties. Some related studies on notable silver coordination polymers [Antimicrob. Agents and Chemotherapy 2010, 54, 4208; Inorg. Chem. 2016, 55, 5886; J. Inorg. Biochem. 2014, 138, 114; CrystEngComm, 2013, 15, 8060] are suggested to be referenced.
5. Also, it would be good to comment in the conculsions that such a type of Ag-pyrazolate compounds might be interesting to explore for their potential antimicrobial properties. In addition, a couple of related books on bioactive metal-based agents can be cited [Antimicrobial Materials for Biomedical Applications (S. Farah, K. Reddy Kunduru, A.J. Domb), RSC, 2019; Bio- and Bioinspired Nanomaterials, Eds. D. Ruiz-Molina, F. Novio, C. Roscini, John Wiley & Sons, 2014].
6. Please check if all the abbreviations are explained in the manuscript.
7. Keywords. Add "coordination polymer" instead of "coordination".
8. Graphical abstract is not supplied (not available to this reviewer).
Round 2
Reviewer 2 Report
There is no flaw in chemistry or experiment. Also well written scientifically as well as language wise to. My only comment is that the work is simple and routine. Ligands are known already, and these are my worries.
Please go ahead with comments or recommendation using other two referees if that is fine with you.